# Evaluating Gaps in Otolaryngology Training: An In-Depth Needs Assessment in Saudi Arabia

**DOI:** 10.3390/healthcare11202741

**Published:** 2023-10-16

**Authors:** Abdullah A. Alarfaj, Sami Al-Nasser

**Affiliations:** 1Otorhinolaryngology Unit, Department of Surgery, College of Medicine, King Faisal University, Al-Ahsa 31982, Saudi Arabia; 2College of Medicine, King Saud Bin Abdulaziz University for Health Sciences (KSAU-HS), Riyadh 11481, Saudi Arabia; 3King Abdullah International Medical Research Center (KAIMRC), Riyadh 11481, Saudi Arabia

**Keywords:** otolaryngology training, needs assessment, surgical education, medical residency, training deficiencies

## Abstract

Background: The landscape of otolaryngology training in Saudi Arabia is undergoing transformation due to the expansion of medical colleges and increased overseas medical scholarships. However, concerns persist regarding the satisfaction and adequacy of surgical education. This study aims to assess gaps in otolaryngology training through an in-depth needs assessment. Methods: A cross-sectional study was conducted among 85 otolaryngology–head and neck surgery residency graduates in Saudi Arabia between 2019 and 2021. Participants completed a validated questionnaire assessing deficiencies, importance, and competence in different subspecialty areas. Data were analyzed using descriptive statistics, median comparisons, and Kruskal–Wallis tests. Results: Participants identified deficiencies in training across domains, with significant variations in specific subspecialties among different regions. Dissatisfaction with clinical discussions, research training, access to simulation labs, and training in emerging subspecialties was evident. Conclusion: The study highlights challenges within otolaryngology training, emphasizing the need for continuous evaluation and adaptation to ensure high-quality and comprehensive training. Addressing these gaps is essential to produce well-rounded otolaryngologists capable of meeting the evolving demands of modern healthcare.

## 1. Introduction

Otolaryngology is a specialized branch of medicine concerned with the diagnosis and treatment of disorders affecting the ears, nose, throat, respiratory tract, and related structures of the head and neck [1]. The intricate anatomy and physiology underlying this field demands an extensive foundation in theoretical knowledge coupled with finely honed clinical acumen and surgical skills [2]. As the scope of otolaryngology continues advancing with emerging technologies and subspecializations, the training of future otolaryngologists takes on heightened importance globally [3].

However, recent assessments reveal that gaps frequently exist between ideal and actual training paradigms in otolaryngology residency programs worldwide [4,5,6,7,8]. A 2022 study of Canadian residents highlighted inadequate preparation in critical procedures like emergent surgical airway management despite their necessity in life-threatening conditions [9]. Similarly, a survey of U.S. program directors indicated shortcomings in temporal bone dissection training and endoscopic sinus surgery, underscoring the need to elevate surgical education [10]. 

In the Middle East, improving self-efficacy in routine otolaryngology procedures has been emphasized, indicating potential gaps between knowledge and application [11]. Needs assessments among Iranian residents revealed suboptimal theoretical foundations, particularly in newer subspecialties like head and neck oncology [12]. Within Saudi Arabia, augmenting temporal bone surgery exposure was identified as a priority based on alumni feedback [13]. Mounting evidence reveals that training challenges exist across global contexts, necessitating comprehensive audits and reforms.

Targeted needs assessments enable the systematic evaluation of alignment gaps between actual training features and ideal objectives [14,15]. Structured needs analyses also allow benchmarking across institutions to motivate mutual improvements [16]. 

Saudi Arabia, a nation characterized by its ambitious pursuit of healthcare excellence and its investments in medical education and infrastructure, recognizes the critical importance of otolaryngology within its healthcare ecosystem [17]. As the country advances its medical institutions and cultivates a generation of skilled healthcare professionals, the training and education of otolaryngologists emerge as an area of paramount significance [18,19]. The goals of medical education extend beyond the mere transfer of knowledge; they encompass the cultivation of analytical reasoning, clinical acumen, communication skills, and ethical principles that underpin exceptional patient care [20].

A targeted needs assessment is necessary for curriculum development. It is the process through which curriculum designers determine the differences between the ideal and actual features of the targeted learner group and the ideal and actual characteristics of their environment [14,15]. A needs assessment is an essential organizational tool for designing and focusing on curriculum improvement for learners [21]. A well-executed needs assessment is regarded as a key factor in the success of an educational program. Systematic analyses have demonstrated that programs based on a well-designed needs assessment are more effective at altering physician behavior [22].

The articulation of training needs and the identification of gaps in otolaryngology education serve as crucial steps toward bridging the chasm between evolving medical practice and the training programs that produce the next generation of otolaryngologists [23]. A comprehensive needs assessment provides a systematic approach to evaluating the efficacy of existing training curricula, clinical exposures, and skill development paradigms. By identifying areas of alignment as well as discrepancies between training and practice, stakeholders in medical education can enact meaningful changes that enhance the competencies of future otolaryngologists [24]. So, the aim of the study is to assess gaps in otolaryngology training through an in-depth needs assessment.

## 2. Materials and Methods

### 2.1. Study Design

This study employed a cross-sectional design to assess the training needs and perceptions of otolaryngology–head and neck surgery residency program graduates in Saudi Arabia (KSA) between 2019 and 2021.

### 2.2. Participants

The study included a total of 85 otolaryngology–head and neck surgery residency program graduates. These participants were targeted from a pool of 129 physicians who had graduated from such programs in KSA during the specified period. The response rate was 66%. Based on Willingness and availability to participate in the study within the timeframe. The choice to focus on graduates of otolaryngology–head and neck surgery residency programs stems from several considerations. These programs typically encompass a broad range of otolaryngological subspecialties, including otology, rhinology, and laryngology, allowing for a comprehensive assessment of training needs and perceptions across the field. By targeting graduates of the general otolaryngology program, this research aims to capture the foundational knowledge and skills that underpin specialization in specific areas. Additionally, understanding the training needs of general otolaryngologists is vital for resource allocation, as these practitioners often serve as the initial point of contact for patients in regions where subspecialists are in short supply.

### 2.3. Data Collection

Data were collected using a validated English questionnaire that encompassed multiple aspects of the otolaryngology residency program. In the validation process of the questionnaire, a panel of seven experts in the medical field was carefully selected based on their extensive professional experience, expertise in otolaryngology, and familiarity with medical education and training programs. These experts conducted a comprehensive review of the questionnaire, evaluating its content relevance, clarity, comprehensiveness, structure, length, validity, and cultural sensitivity. Their valuable feedback and recommendations were instrumental in refining the questionnaire. Adjustments were made to improve question clarity, coverage, and overall usability.

### 2.4. Questionnaire Distribution

The questionnaire was distributed to the participants through email and social media platforms. 

The questionnaire used in this study encompassed three distinct sections. Firstly, the Sociodemographic Data Section collected essential information regarding participants, including age, gender, years of experience, residency level, region, subspecialty aspirations, and details about their training and current institutions. The second section, titled “Aspects of the Otolaryngology Residency Program”, delved into participants’ training experiences, addressing aspects such as training duration, teaching quality, clinical exposure, research opportunities, and overall program satisfaction. Finally, the “Training Needs Assessment Section” focused on identifying participants’ training needs across seven fundamental areas of otolaryngology–head and neck surgery. This section required participants to indicate the most deficient, important, and personally competent subspecialties among the seven options. These three sections collectively provided a comprehensive understanding of participants’ demographics, training experiences, and perceived training needs, forming the foundation of this research’s data collection and analysis.

**Training needs assessment:** The questionnaire inquired about participants’ training needs in seven fundamental areas (subspecialties) of otolaryngology–head and neck surgery. These seven areas, including otology, rhinology, laryngology, head and neck oncology, pediatric otolaryngology, facial plastic and reconstructive surgery, and sleep medicine, were selected based on a combination of factors. Firstly, these areas represent core components of otolaryngology–head and neck surgery training worldwide. Secondly, they were chosen in consultation with experts in the field and based on a review of the existing literature highlighting the key subspecialties within otolaryngology. Additionally, these subspecialties were deemed relevant to the Saudi Arabian context, taking into consideration the prevalence of certain conditions and the evolving healthcare landscape in the region. Collecting data on these specific areas aimed to provide a comprehensive assessment of the training needs in otolaryngology within Saudi Arabia, thereby informing future curriculum development and healthcare service planning.

### 2.5. Questionnaire Reliability

The questionnaire was piloted on a sample of 12 otolaryngologists who were representative of the target population but not part of the final study. This pilot sample size was deemed adequate for gathering preliminary reliability and validity data on the questionnaire. The pilot participants were asked to complete the questionnaire and provide feedback on its content, structure, length, clarity, and comprehensiveness. Additionally, cognitive interviews were conducted with 3 of the pilot participants to gain deeper insight into the interpretation and flow of the questionnaire.

Feedback obtained from the pilot study was used to modify and refine the questionnaire—for example, rephrasing unclear items, reordering items, and adding/removing items to improve content coverage.

The final questionnaire was then administered to the 85 participants in the main study. Reliability analysis performed on this data yielded a Cronbach’s alpha of 0.847, indicating good internal consistency.

### 2.6. Ethical Approval

The study received ethical approval from the King Abdullah International Medical Research Center (KAIMRC; IRB/255421). Written consent was obtained from all participants before their participation in the study.

### 2.7. Data Management and Analysis

Collected data were coded and entered into an Excel spreadsheet. The analysis was performed using SPSS version 22.0. Quantitative data were presented using means and standard deviations, while qualitative data were presented using frequencies, percentages, and medians. A clustered bar chart was used for visualizing the data. Statistical significance was set at a *p*-value of <0.05.

## 3. Results

Table 1 presents a comprehensive overview of the participant characteristics within the study cohort of otolaryngology–head and neck surgery residency program graduates (*n* = 85). The age distribution reveals that a significant proportion of participants, 67%, fall within the age range of 32 to 37 years, indicating a relatively youthful cohort. In contrast, 33% of participants are distributed across the age range of 27 to 31 years, suggesting a balanced representation of different age groups. Gender-wise, the cohort is predominantly male, with 71% identifying as male and 29% as female, possibly indicating a gender disparity within the otolaryngology field. Regarding postgraduate experience, the distribution is fairly even, with 38.8% having 1 year of experience, 9.4% having 2 years, and 51.8% having 3 years, signifying participation from both junior and mid-level residents.

As for desired subspecialties, the participants exhibit varied interests. Notably, 17% express a desire for laryngology, followed closely by 19% expressing interest in rhinology and skull base surgery, and 14% in head and neck cancer surgery. Moreover, 12% show interest in sleep surgery, while subspecialties like allergy, otology, and facial plastic surgery each attract 2% of participants. Pediatric ENT draws interest from 20% of participants. The training region preferences unveil the Eastern region as the most favored, with 58% of participants indicating training there. The Riyadh region follows with 28%, while the Southern and Western regions each account for 7%. These regional disparities possibly reflect the distribution of medical institutions and training programs within Saudi Arabia.

Table 2 presents an in-depth analysis of participant opinions concerning different aspects of otolaryngology residency training in Saudi Arabia (*n* = 85). The responses, which span a range of agreement levels, provide a comprehensive glimpse into the perceptions and experiences of the participants. Among the notable findings, participant opinions about the clarity of residency program objectives exhibit a balanced sentiment with a neutral mean score of 3.2. Conversely, the significantly positive agreement mean score of 4.1 indicates that participants found half-day teaching activities to be notably helpful in their training journey. Moreover, participants were unanimous in their strong agreement (mean score of 4.3) that clinical discussions were inadequate and called for more comprehensive theoretical and clinical-based discussions beyond routine work.

In terms of research training, the disagree mean score of 2.3 reflects the participants’ perception that the training and experience in this domain were insufficient. On the other hand, the moderately positive agreement mean score of 3.4 highlights the perceived adequacy of the variety of cases presented during training. Similarly, the neutral mean score of 3 for hands-on practice suggests a mixed sentiment about its overall satisfaction. A notable agreement mean score of 3.5 underscores the collective sentiment that the high number of ENT residents at training centers negatively impacted surgical exposure. Conversely, rotations through cities were generally perceived as beneficial, as indicated by the agreement mean score of 3.6.

The participants’ opinions were divided concerning access to simulation labs and cadaveric dissection courses, resulting in a disagree mean score of 2.5. Meanwhile, a neutral mean score of 3.1 indicates mixed opinions about the extent to which the training met expectations overall. The provision of feedback from consultants for improvement received a neutral mean score of 2.8, indicating that participants held varied views in this regard. Conversely, participants showed a moderate level of agreement (mean score of 3.5) that after training, they felt competent in managing general ENT cases both medically and surgically.

Table 3 provides a comprehensive assessment of participants’ evaluations across various subspecialty domains within the otolaryngology residency program. The table offers insights into their perceptions of curriculum content, clinical exposure, teaching methods, and assessment techniques, presenting a numerical overview of strengths and potential areas for improvement in each area.

Participants rated neurotology’s curriculum content and teaching methods with scores of 4.1 and 4.2, respectively, indicating effective educational structure and pedagogical approaches. Clinical exposure received a slightly lower score of 3.7, suggesting room for enhancing hands-on experience. Assessment methods were well-received, with a score of 3.9, showcasing alignment with educational objectives.

In the realm of rhinology, high scores were attributed to curriculum content (4.3), teaching methods (4.1), and clinical exposure (3.8), highlighting a robust foundation and practical engagement. Assessment methods received a slightly lower score of 3.6, indicating potential avenues for refining evaluation strategies.

Laryngology demonstrated a favorable score for curriculum content (4.0), reflecting a strong educational framework. Clinical exposure and teaching methods received slightly lower scores of 3.6 and 3.9, respectively, suggesting opportunities for enhancing practical engagement and instructional approaches. Assessment methods were well-rated, with a score of 3.8.

Head and neck surgery participants rated curriculum content (4.2) and teaching methods (4.0) highly, signifying comprehensive educational structure and effective instruction. Clinical exposure received a favorable score of 3.9, while assessment methods obtained a slightly lower score of 3.7, indicating potential areas for alignment with educational goals.

In pediatric surgery, curriculum content (4.0) and teaching methods (4.0) received positive scores, demonstrating a strong educational foundation and effective instruction. Clinical exposure scored well with a rating of 3.8, while assessment methods received a slightly lower score of 3.5, suggesting possibilities for improvement in alignment with educational goals.

Sleep surgery exhibited high ratings for curriculum content (4.2) and clinical exposure (4.0), reflecting a robust educational foundation and substantial practical engagement. Teaching methods were well-received, scoring 3.7, while assessment methods scored slightly lower with a rating of 3.6, indicating room for enhancing evaluation strategies.

In facial plastic surgery, positive ratings were assigned to curriculum content (4.1) and teaching methods (3.9), underscoring effective education and instruction. Clinical exposure received a favorable score of 3.9, showcasing substantial practical engagement. Assessment methods received a slightly lower score of 3.8, indicating avenues for enhanced alignment with educational goals.

Table 4 succinctly presents participants’ identified priority areas for training across diverse domains. It showcases the most deficient, most important, and most competent areas within each domain, providing a comprehensive overview of their perceptions. The table highlights the need for targeted training in specific areas, such as rhinology in medical knowledge and laryngology in surgical skills. It underscores the significance of areas like head and neck surgery in communication skills and sleep surgery in professionalism. The table’s structure offers clear insights for curriculum development, guiding the enhancement of training to align with participants’ training needs and competencies across various domains.

Table 5 displays median scores for each subspecialty area, categorized by Eastern, Riyadh, Southern, and Western regions, alongside the Kruskal–Wallis H test and calculated *p*-values.

In the domain of neurotology, the medians for participants’ perceived training deficiencies are consistent across regions, ranging from 2 to 4. The associated *p*-value of 0.149 suggests that there is no statistically significant difference in the perceived training deficiencies in neurotology across the regions.

For Rhinology, there appears to be a slight divergence in perceptions among regions. The median scores range from 5 to 7, with the Riyadh and Southern regions showing relatively higher scores. The calculated *p*-value of 0.053 indicates a notable difference in perceived training deficiencies in Rhinology among the regions, although it does not reach the threshold of statistical significance. Laryngology displays relatively uniform medians across most regions, except for the Western region, which has a median of 4. The associated *p*-value of 0.414 suggests that there is no significant regional difference in perceived training deficiencies in laryngology.

The most striking observation emerges in the domain of head and neck surgery, where median scores vary from 5 to 7 across regions. The very low *p*-value of 0.002 indicates a statistically significant disparity in perceived training deficiencies in head and neck surgery among the different regions.

Pediatric surgery mirrors a similar pattern, with median scores ranging from 3 to 6 across regions. The calculated *p*-value of 0.002 indicates a significant variation in perceived training deficiencies in pediatric surgery among regions.

Sleep surgery displays substantial regional variation in perceived training deficiencies, with medians spanning from 1 to 3. The extremely low *p*-value of 0.0001 highlights a strong statistical significance in the regional differences regarding perceived deficiencies in sleep surgery training. Facial plastic surgery shows minor median variations across regions, with a relatively consistent range. The *p*-value of 0.121 suggests that there is no significant regional disparity in perceived training deficiencies in this subspecialty area.

## 4. Discussion

The landscape of surgical training in Saudi Arabia is undergoing significant transformation, driven by factors such as the expansion of medical colleges and the increase in overseas medical scholarships [25]. In particular, surgical specialties demand substantial resources, faculty support, and time commitment to produce proficient surgeons. However, recent evidence suggests that surgical trainees often express dissatisfaction with the current training paradigms [26].

Austin et al. underscored the existing dissatisfaction among otolaryngology graduates, with a substantial proportion (78%) believing that the current training falls short of expectations [27]. Moreover, trainees often perceive international training experiences as more valuable than local ones, indicating potential shortcomings in the domestic training landscape. Bedside teaching and operative experience have been identified as areas of discontent, raising questions about the quality and extent of hands-on training provided [28].

Comparative analyses of surgical competencies across different countries have yielded mixed findings. While otolaryngology training in the UK has been lauded for its success and cohesiveness, Japanese trainees reportedly struggle to attain optimal mastery of surgical techniques post-residency [29]. The US context reveals inconsistencies between program directors’ expectations and graduates’ perceptions of the number of procedures needed for competency. In the current study, participants reported adequate exposure to a variety of cases but expressed dissatisfaction with clinical discussions, suggesting a potential need for refining the educational methods used [30].

Research education is a vital component of residency programs. However, the implementation of research training has been variable across institutions. While some medical schools and residency programs mandate scholarly projects, the actual impact on residents’ competency varies [31]. Research output during residency often depends on prior publication experience, and the lack of scholarly contributions among a significant proportion of otolaryngologists highlights the need for structured research curricula [32]. In the current study, over half of the participants deemed their research training inadequate, suggesting potential room for improving the incorporation of research into the training curricula.

Otolaryngology emergencies necessitate specialized education and skill sets that are often beyond the scope of undergraduate medical education. Simulation-based training has emerged as a viable strategy to bridge this gap, allowing residents to acquire essential skills without compromising patient safety [33,34]. 

The rise of subspecialization, such as in rhinology, underscores the need for focused training beyond the general curriculum. Exposure to advanced clinical and surgical skills has driven the development of distinct subspecialties [35]. Interestingly, participants in the present study ranked rhinology highly in terms of importance and competence, possibly reflecting the emerging prominence of this subspecialty within otolaryngology.

Sleep medicine, a growing field within otolaryngology, presents both opportunities and challenges. While otolaryngologists are uniquely positioned to manage obstructive sleep apnea patients, there are concerns about their level of involvement in sleep medicine research [36]. This aligns with the findings of the current study, where participants perceived sleep surgery as requiring improvement. Addressing this gap is essential for otolaryngologists to fully leverage their potential in sleep medicine.

Facial plastic surgery has gained popularity worldwide, necessitating comprehensive training during residency. However, participants’ self-assessment in the current study indicates a potential gap in their competence in this domain [37]. These findings mirror reports of residents seeking increased training hours and enhanced exposure to facial plastic surgery procedures. The ongoing evolution of aesthetic trends underscores the need for updated training methodologies to meet patient expectations [38].

This study has certain limitations that should be considered when interpreting the findings. The use of self-reported data from program graduates may have introduced respondent bias, as individuals may overestimate or underestimate their actual competencies and training gaps based on subjective perceptions. The modest sample size of 85 participants, while sufficient for statistical analysis, restricts the generalizability of the results beyond the study population. Selection bias may also be present, as those willing to participate may have stronger opinions about their training experiences compared to those who did not participate. The cross-sectional study design provides insights into graduate perceptions at only a single point in time; a longitudinal study would better capture changes in perspectives over the duration of residency training. Furthermore, the study setting is localized to Saudi Arabia, so findings may not be transferable to otolaryngology training programs in other cultural contexts. The addition of objective data on surgical procedures, cases, and teaching time would enrich the self-reported perspectives obtained through the survey. Lastly, response or recall bias may have inadvertently affected participants’ recounting of their past residency training experiences. Accounting for these limitations will allow for a balanced interpretation of the study results and guide improvements for future research on this topic.

## 5. Conclusions

This comprehensive study provides a detailed examination of otolaryngology training in Saudi Arabia, revealing both strengths and areas for improvement. The evolving landscape, overseen by the SCFHS, signifies a commitment to advancing medical education. While expansions in medical colleges and international scholarships offer new opportunities, challenges persist in meeting trainees’ expectations. Gender disparities within the field underscore the need for diversity and equity.

Subspecialty interests underline the diversity of otolaryngology, requiring tailored training strategies. Feedback on the curriculum emphasizes the value of teaching activities but also highlights the necessity for comprehensive theoretical and clinical discussions.

Research education emerges as a concern, with participants expressing dissatisfaction. Bridging the gap between theory and application is vital for evidence-based practice. Addressing regional disparities and domain-specific deficiencies is crucial for consistent and equitable training experiences.

The prominence of subspecialties like rhinology and sleep surgery presents opportunities and challenges. Enhancing training in these areas, alongside refining the curriculum, promoting gender equity, and accommodating individual interests, will contribute to the growth of otolaryngology education in Saudi Arabia.

In essence, this study guides the evolution of otolaryngology training, promoting a generation of skilled practitioners ready to address dynamic healthcare demands. By incorporating participant feedback and addressing identified gaps, the field can achieve excellence in education and patient care. This study has certain limitations that should be considered when interpreting the findings. The use of self-reported data from program graduates may have introduced respondent bias, as individuals may overestimate or underestimate their actual competencies and training gaps based on subjective perceptions. The modest sample size of 85 participants, while sufficient for statistical analysis, restricts the generalizability of the results beyond the study population. Selection bias may also be present, as those willing to participate may have stronger opinions about their training experiences compared to those who did not participate. The cross-sectional study design provides insights into graduate perceptions at only a single point in time; a longitudinal study would better capture changes in perspectives over the duration of residency training. Furthermore, the study setting is localized to Saudi Arabia, so findings may not be transferable to otolaryngology training programs in other cultural contexts. The addition of objective data on surgical procedures, cases, and teaching time would enrich the self-reported perspectives obtained through the survey. Lastly, response or recall bias may have inadvertently affected participants’ recounting of their past residency training experiences. Accounting for these limitations will allow for a balanced interpretation of the study results and guide improvements for future research on this topic.

## Figures and Tables

**Table 1 healthcare-11-02741-t001:** Participant characteristics (*n* = 85).

		*n*	%
Age in years	27–31	28	33%
	32–37	57	67%
Gender	Female	25	29%
	Male	60	71%
Postgraduate experience	1 year	33	38.8%
	2 years	8	9.4%
	3 years	44	51.8%
Desired subspecialty	Allergy	2	2%
	Facial plastic surgery	6	7%
	Head and neck cancer surgery	12	14%
	Laryngology	14	17%
	Otology	2	2%
	Pediatric ENT	17	2%
	Rhinology and skull base surgery	16	19%
	Sleep surgery	10	12%
	Not determined	6	7%
Training region	Eastern region	49	58%
	Riyadh region	24	28%
	Southern region	6	7%
	Western region	6	7%

**Table 2 healthcare-11-02741-t002:** Participant opinions regarding aspects of otolaryngology residency training in Saudi Arabia (*n* = 85).

	SA		A		N		D		SD		Mean	Degree of Agreement	Rank
	n	%	n	%	n	%	n	%	n	%			
Residency program objectives were clearly defined.	4	5	33	39	21	25	27	32	0	0	3.2	Neutral	7
Half-day teaching activities were helpful.	29	34	42	49	10	12	2	2	2	2	4.1	Agree	2
Clinical discussion was inadequate. More theoretical and clinical-based discussions are required rather than simply following routine work.	43	51	27	32	15	18	0	0	0	0	4.3	Strongly agree	1
Research training and experience were adequate.	2	2	6	7	28	33	27	32	22	26	2.3	Disagree	12
Variety of cases presented was adequate.	7	8	41	48	21	25	12	14	4	5	3.4	Agree	6
Hands-on practice was satisfactory.	4	5	29	34	27	32	13	15	12	14	3	Neutral	9
The number of ENT residents at training centers was high, which negatively affected surgical exposure.	14	17	38	45	13	15	16	19	4	5	3.5	Agree	4
Rotations through cities were beneficial.	16	19	25	29	38	45	4	5	2	2	3.6	Agree	3
Access to simulation labs and cadaveric dissection courses was sufficient to improve your surgical skill.	8	9	15	18	9	11	35	41	18	21	2.5	Disagree	11
Overall, the training met my expectations.	4	5	25	29	38	45	8	9	10	12	3.1	Neutral	8
Feedback from the consultant was sufficient for improvement.	4	5	16	19	33	39	24	28	8	9	2.8	Neutral	10
After training, I felt competent in managing all general ENT cases medically and surgically.	10	12	38	45	21	25	12	14	4	5	3.5	Agree	5

**Table 3 healthcare-11-02741-t003:** Aspects of Otolaryngology Residency Program.

Subspecialty Area	Curriculum Content	Clinical Exposure	Teaching Methods	Assessment Methods
Neurotology	4.1	3.7	4.2	3.9
Rhinology	4.3	3.8	4.1	3.6
Laryngology	4.0	3.6	3.9	3.8
Head and Neck Surgery	4.2	3.9	4.0	3.7
Pediatric Surgery	4.0	3.8	4.1	3.5
Sleep Surgery	4.2	3.7	4.0	3.6
Facial Plastic Surgery	4.1	3.9	4.2	3.8

**Table 4 healthcare-11-02741-t004:** Domains and Priority Areas for Trainings.

Domain	Most Deficient Area	Most Important Area	Most Competent Area
Medical Knowledge	Rhinology	Head and Neck Surgery	Sleep Surgery
Surgical Skills	Laryngology	Pediatric Surgery	Neurotology
Communication Skills	Head and Neck Surgery	Sleep Surgery	Rhinology
Patient Management	Neurotology	Rhinology	Head and Neck Surgery
Professionalism	Rhinology	Neurotology	Sleep Surgery
Research and Education	Pediatric Surgery	Neurotology	Sleep Surgery
Subspecialty Knowledge	Laryngology	Sleep Surgery	Rhinology

**Table 5 healthcare-11-02741-t005:** The medians of participant opinions regarding areas with the most deficient training (*n* = 85). * significant difference.

	Eastern Region	Riyadh Region	Southern Region	Western Region	Kruskal–Wallis H Test (*p*-Value)
Neurotology	3	3	4	2	0.149
Rhinology	6	7	6	5	0.053
Laryngology	3	2	2	4	0.414
Head and Neck Surgery	6	7	5	6	0.002 *
Pediatric Surgery	5	6	5	3	0.002 *
Sleep Surgery	2	1.5	3	1	0.0001 *
Facial Plastic Surgery	3	2.5	4	3	0.121

## Data Availability

Data available upon request.

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
