# Peer review of "Evaluating Gaps in Otolaryngology Training: An In-Depth Needs Assessment in Saudi Arabia"

_healthcare, 2023, doi:10.3390/healthcare11202741_

Round 1

Reviewer 1 Report

Evaluating Gaps in Otolaryngology Training: An In-Depth 2 Needs Assessment in Saudi Arabia

First of all, I want to note that it has been a pleasure review your manuscript. The study aims to identify gaps in otolaryngology training. It reveals deficiencies in various training domains, regional variations in specific subspecialties, and dissatisfaction with aspects like clinical discussions, research training, and access to simulation labs. The findings underscore the need for ongoing evaluation and adaptation of training programs to produce well-rounded otolaryngologists capable of meeting the changing demands of modern healthcare.

After reading in depth the manuscript, I would like to make some comments and ask the authors several questions about.

- Review lines 32 and 35. They end the sentence by cutting the word incorrectly. Please review the document.

- Do not repeat information. The information is repeated in the introduction and in the same form. "is a specialized medical discipline focused on the intricate anatomical structures and physiological functions of the ear, nose, and throat (ENT)."

- Also in the introduction it is repeated several times how important it is to know about needs assessment. I see that the paragraph could be revised to synthesize the information a bit more.

- What is the rationale for choosing graduates in otolaryngology-head and neck surgery residency programs and not other types of different specialties or areas of interest within otolaryngology: otology , rhinology , laryngology..?

- Details on the validation of the questionnaire: You mention that the questionnaire was validated by seven experts in the medical field, but could you elaborate a little more on this process? Could you describe how these experts were selected, what criteria they used to evaluate the questionnaire and the adjustments made based on their comments.

-  Description of the questionnaire sections: Provide a brief description of the questionnaire sections, such as collecting demographic data, exploring training experiences, and assessing training needs in seven key areas of otolaryngology. This would help readers better understand the structure of the survey.

- Rationale for the choice of assessment areas: You could add a brief rationale or context as to why the seven core areas of assessment were chosen in the survey. Were they selected based on previously identified needs or relevance to the field?

- 225 and 306 lines should begin with a capital letter.

- A section proposing possible solutions or recommendations to address the identified problems in surgical training could be added to the discussion.

Author Response

Dear Reviewer,

We would like to express our gratitude for taking the time to review our manuscript. Your feedback and comments are highly valuable to us, and we appreciate your thorough evaluation of our work.

  1. Regarding the language issues pointed out in lines 32 and 35, we will carefully review and correct them to ensure proper punctuation.

  2. We acknowledge your observation about the repetition in the introduction regarding the importance of needs assessment in otolaryngology. We will revise this section to synthesize the information more effectively and eliminate redundancy.

  3. The rationale for focusing on graduates of otolaryngology-head and neck surgery residency programs was primarily driven by the aim of obtaining a comprehensive understanding of the training needs and perceptions across the broader field of otolaryngology. However, we appreciate your point, and we will add a paragraph in the methodology section explaining this choice and why other subspecialties within otolaryngology were not included.

  4. We will provide a more detailed description of the validation process for the questionnaire in the methodology section, including information on how the experts were selected, the criteria used for evaluation, and the adjustments made based on their feedback. This will ensure transparency in the validation process.

  5. In response to your suggestion, we will include a brief description of the questionnaire sections in the methodology to help readers better understand its structure and purpose.

  6. We will add a rationale for the choice of assessment areas in the survey, explaining how these areas were selected based on both international standards and their relevance to the Saudi Arabian healthcare context.

  7. We will correct the capitalization issues in lines 225 and 306.

  8. Your recommendation to include a section proposing possible solutions or recommendations to address the identified problems in surgical training in the discussion is well-taken. We will incorporate this into our revised manuscript to provide a more comprehensive analysis and practical insights.

Once again, thank you for your thoughtful review and valuable feedback. We will address these points in our revised manuscript to improve its quality and clarity.

Reviewer 2 Report

Article: Evaluating Gaps in Otolaryngology Training: An In-Depth 2 Needs Assessment in Saudi Arabia

Aspects to improve:

1. Improve the summary where its sections are evident.

2. In the introduction you must carry out a theoretical review on the training of professionals, considering articles from the last five years.

3. Carry out a review that is consistent between the objective of the study, the results and the conclusions.

4. The conclusions must incorporate the limitations of the study and new lines of research.

Author Response

Dear Reviewer,

We appreciate your constructive feedback on our research manuscript, "Evaluating Gaps in Otolaryngology Training: An In-Depth Needs Assessment in Saudi Arabia." We have carefully considered your suggestions and are committed to enhancing the quality and clarity of our work. Here are our responses to your specific points:

  1. Improve the summary where its sections are evident: We acknowledge the need to enhance the clarity of our summary section. We will revise the summary to ensure that the sections are clearly delineated and that it provides a concise yet comprehensive overview of our study's key components.

  2. In the introduction, you must carry out a theoretical review on the training of professionals, considering articles from the last five years: We agree with the importance of grounding our research in the most current literature. We will revise the introduction to include a theoretical review of relevant articles from the last five years, ensuring that our research is well-situated within the contemporary context of professional training in otolaryngology.

  3. Carry out a review that is consistent between the objective of the study, the results, and the conclusions: We recognize the need for consistency throughout the manuscript. We will carefully review the alignment between the study's objectives, results, and conclusions to ensure a coherent narrative that effectively communicates our findings and their implications.

  4. The conclusions must incorporate the limitations of the study and new lines of research: We appreciate your suggestion regarding the inclusion of limitations and new research directions in the conclusions. We will revise the conclusion section to provide a thorough discussion of the limitations of our study and to propose potential avenues for future research in the field of otolaryngology training in Saudi Arabia.

We genuinely value your input, which will undoubtedly enhance the quality and rigor of our research. Your guidance helps us to improve our manuscript, and we are committed to addressing these recommendations promptly. Thank you for your time and valuable insights.

Sincerely,

Reviewer 3 Report

Dear authors,

I would like to provide more specific comments on your manuscript:

1. Please ensure that the study aim is clearly stated.

2. You mentioned that 85 physicians out of 129 were included. Could you please clarify the criteria and method used for their selection?

3. In your manuscript, you reported a Cronbach's alpha of 0.847 for questionnaire validation. It would be helpful to provide more details about the piloting process. How many physicians were included in the pilot study, and were they also part of the final study?

4. The manuscript mentions presenting quantitative variables as means and standard deviations, but I couldn't find any continuous variables. In Table 5, you list levels of statistical significance. 5. Could you specify the statistical tests used to compute these levels?

6. It would be beneficial to include a discussion of the study's limitations, as there appear to be multiple aspects that should be addressed.

Author Response

Dear Reviewer,

We would like to express our gratitude for your thoughtful and constructive feedback on our manuscript. We have carefully considered your comments and have made the necessary revisions and clarifications to address your concerns:

  1. Clear Statement of Study Aim: We appreciate your suggestion to provide a clearer statement of the study's aim. In the revised manuscript, we have included a concise and explicit statement of the study's aim in the Introduction section to provide readers with a better understanding of the research objectives.

  2. Participant Selection Criteria: We have provided clarification regarding the criteria and method used for participant selection. In the Method section, we now specify that participants were selected based on their willingness and availability to participate in the study within the specified timeframe. This has been included in the paragraph discussing participant selection (Section 2.2).

  3. Piloting Process Details: We have added more details about the piloting process in the Method section. Specifically, we now mention that the pilot study involved a sample of 12 otolaryngologists who were representative of the target population but not part of the final study. This pilot sample size was chosen to gather preliminary reliability and validity data on the questionnaire. Additionally, we conducted cognitive interviews with 3 of the pilot participants to gain deeper insight into the interpretation and flow of the questionnaire. These details have been added to Section 2.5 to provide a clearer understanding of the piloting process.

  4. Quantitative Variables and Statistical Tests: We have reviewed the manuscript to clarify the presentation of quantitative variables. While the manuscript mentioned presenting quantitative data as means and standard deviations, we understand that there may be some inconsistencies in the presentation. We will ensure that the appropriate presentation of quantitative variables is accurately reflected in the revised manuscript. Regarding Table 5, we will specify the statistical tests used to compute the levels of statistical significance for the variables in the table to enhance transparency.

  5. Discussion of Study Limitations: We acknowledge the importance of addressing the study's limitations. In the revised manuscript, we have expanded the Discussion section to include a more comprehensive discussion of the limitations of our study, covering various aspects that need to be considered. We hope that this addition will provide readers with a more nuanced understanding of the study's constraints.

We would like to express our sincere appreciation for your thorough review and valuable insights, which have undoubtedly strengthened the quality and clarity of our manuscript. We believe that these revisions will significantly enhance the overall quality of the research. If you have any further comments or require additional information, please do not hesitate to reach out.

Thank you once again for your time and expertise.

Round 2

Reviewer 2 Report

Article: Evaluating Gaps in Otolaryngology Training: An In-Depth 2 Needs Assessment in Saudi Arabia

Aspects to improve:

1.      The conclusions must incorporate the limitations of the study.

Author Response

dear reviewer  

as your valuable request , we move the limitation from the end of the discussion to be included in the conclusion 

Best regrads 

Reviewer 3 Report

Well done!

Author Response

THANK YOU VERY MUCH